# Trends in Open vs. Endoscopic Carpal Tunnel Release: A Comprehensive Survey in Japan

**DOI:** 10.3390/jcm11174966

**Published:** 2022-08-24

**Authors:** Michiro Yamamoto, James Curley, Hitoshi Hirata

**Affiliations:** Department of Hand Surgery, Nagoya University Graduate School of Medicine, 65 Tsurumai-cho, Showa-ku, Nagoya 466-8550, Japan; curley@med.nagoya-u.ac.jp (J.C.); h-hirata@med.nagoya-u.ac.jp (H.H.)

**Keywords:** carpal tunnel syndrome, carpal tunnel release, trends, Japan

## Abstract

We analyzed trends in open and endoscopic carpal tunnel release (CTR) from 2014 to 2019 using the National Database of Health Insurance Claims and Specific Health Checkups in Japan (NDB). Japan has a universal health insurance system and more than 95% of all claims are searchable in the NDB open data repository. The results revealed that nearly 40,000 CTRs were performed annually in Japan, and open CTR was performed almost 4 times more often than endoscopic CTR. The crude annual incidence of CTR in the general population among people 20 years of age or older was 32.2 per 100,000. The incidence of open CTR peaked in the 80–84 age range for both males and females. The incidence of endoscopic CTR peaked at 80–84 years in females and at 75–79 years in males. There was a mild correlation coefficient between the endoscopic CTRs and the number of hand surgery specialists by prefecture per population (*r* = 0.32, *p* = 0.04). However, the number of hand surgeons per capita by region and open CTR per capita was not correlated (*r* = 0.06, *p* = 0.67). There were about twice as many outpatient as inpatient surgeries, reflecting a trend toward ambulatory treatment.

## 1. Introduction

Carpal tunnel syndrome (CTS) is the most common compression neuropathy. In the general population, one in five symptomatic individuals can be expected to have CTS based on a clinical and electrophysiological examination [1]. The prevalence of CTS using different case definitions ranges from 2.5 to 11.0% [2]. Open carpal tunnel release (OCTR) and endoscopic carpal tunnel release (ECTR) have been performed for symptomatic patients with successful results after failed conservative treatments such as splinting, medications, and corticosteroid injections [3]. Both procedures are widely utilized and there are no significant differences regarding the long-term postoperative results [4]. Although ECTR has advantages, such as minimal scarring and a shorter recovery period which facilitates an earlier return to activities of daily life, there are concerns about the potential for transient or permanent nerve injury, and these serious consequences should not be underestimated [5]. The transverse carpal ligament is divided in both OCTR and ECTR, although each technique has its own advantages and disadvantages [6]. The regional distribution of hand surgery specialists may also influence differences in surgical procedures, and while general orthopedic surgeons might perform a percentage of OCTR, endoscopic surgery requires more specialized skills. 

The overall risk and relative severity of CTS increases with age [7]. Carpal tunnel release (CTR) is widely performed in Japan, which has a rapidly aging population [6]. As the leading super-aged society in the world, Japan serves as a demographic bellwether regarding health conditions associated with advanced age and their treatments. According to reports from the United States and Canada, CTR for the elderly is increasing [7,8,9]. Being alert to the trends in Japan associated with the age distribution of patients who have undergone CTR will help other countries that are anticipating becoming super-aged societies themselves to be prepared.

The National Database of Health Insurance Claims and Specific Health Checkups of Japan (NDB) is one of the largest, most comprehensive, national-level healthcare data repositories in the world. It is thorough and contains complete datasets of insured medical care delivered within the country’s universal healthcare system. Since 2014, this information has been parsed, compiled in spreadsheets, and published annually in NDB Open Data Japan (NDB-ODJ). As a result, more than 95% of ECTR and OCTR claims are accessible in the form of anonymized statistics—for example, surgical type, age, and geographic location—drawn from health insurance claims [10]. 

The purpose of this study was to document the annual OCTR and ECTR figures within Japan for procedures conducted between 2014 and 2019 and to analyze their trends, gender differences, age distributions, and regional variations based on this comprehensive survey.

## 2. Materials and Methods

In this study, the designation “NDB” refers to the National Database of Health Insurance Claims and Specific Health Checkups of Japan administered by the Ministry of Health, Labour and Welfare, while “NDB-ODJ” refers to NDB Open Data Japan published as spreadsheets that summarize the claims’ data. For 2014–2019, we accessed the NDB-ODJ site and downloaded the following Excel files: “Number of calculations by division, sex, and age group” and “Number of calculations by prefecture” under “operation (code K)” [11,12,13,14,15,16]. The NDB contains almost all health insurance claims and specific health checkup data associated with the national health insurance system.

In the database, surgical procedures for CTS were classified as either OCTR (K093) or ECTR (K093-2). Information on age, gender, and prefecture was obtained in addition to treatment location, from which a distinction was made between inpatient and outpatient surgeries from 2015 onward.

We characterized the information as follows: (1) *Trends in surgical procedures:* The total number of OCTR and ECTR by year was summarized. Trends and differences, if any, were investigated. (2) *Age distribution by type of surgery and gender:* The mean number of operations from 2014 to 2019 was calculated by surgery type and gender according to age. The age group that underwent surgery most often was investigated. (3) *Age- and sex-specific incidence of OCTR and ECTR:* The crude mean annual incidence of OCTR and ECTR by sex and age from 2014 to 2019 was calculated using a 2019 population estimation summary [17]. The WHO World Standard Population distribution was used to make age-based international comparisons [18]. Calculations using a direct approach were performed to adjust for the age at carpal tunnel release, the Japanese population in 2019, and the crude mean annual incidence of CTR for 6 years. (4) *Number of surgeries by prefecture per population:* The average number of surgeries per 100,000 people in each prefecture from 2014 to 2019 was summarized using a 2019 population estimation summary [17]. The standardized incidence ratio was calculated using the mean cases in each prefecture and the 2019 demographic data [17]. The correlation coefficients between ECTRs or OCTRs and the hand surgery specialists by prefecture per population were analyzed. (5) *Trends in outpatient and inpatient surgeries:* The number of outpatient and inpatient settings by surgical type was compared. 

### Statistical Analysis

An *m* × *n* contingency table was used to determine the differences between the procedures by year. We used the χ-square test to identify the differences in CTR between men and women and the differences in inpatient and outpatient surgery depending on the procedure used. We examined the correlation coefficients of the number of ECTRs and OCTRs and the hand surgery specialists as of 2022 by prefecture per population using the Pearson correlation coefficient. The statistical analysis was performed using the Statcel2 (OMS Publishing, Saitama, Japan) software add-in for Microsoft Excel (Microsoft 365, Microsoft, Redmond, WA, USA). *p* values < 0.05 were considered statistically significant.

## 3. Results

Trends in surgical procedures.

The trends in surgical techniques for the 6-year period beginning in 2014 are shown in Figure 1. There was no significant change in the proportion of surgical procedures over the period (*p* = 0.13). From 2014 to 2019, the ECTR percentages were 28%, 30%, 20%, 21%, 29%, and 30%, respectively, and the overall percentage was 26%.

2.Age distribution by type of surgery and gender.

The mean age distribution by surgical procedure is shown in Figure 2a,b. There was a peak at the 75–79 age range for female patients regardless of the procedure. For male patients, there were peaks at 70–74 years for OCTR and at 65–69 years for ECTR. No difference in CTR was observed for gender (*p* = 0.99).

3.Age- and sex-specific incidence of OCTR and ECTR.

The crude mean annual incidence of OCTR and ECTR by sex and age from 2014 to 2019 is shown in Figure 3a,b. The incidence of OCTR peaked at the 80–84 age range in both males and females. The overall annual incidence of OCTR in the population over the age of 20 per 100,000 was 17.3 (15.6–19, 95% CI) for males and 34.4 (32.5–36.4, 95% CI) for females. The incidence of ECTR peaked at 80–84 years in females and at 75–79 years in males. The overall annual incidence of ECTR in the population over the age of 20 per 100,000 was 4.5 (3.6–5.4, 95% CI) for males and 9 (7.1–10.9, 95% CI) for females. 

The WHO World Standard Population adjustment and the crude annual incidence of CTR per 100,000 people are shown in Figure 3c. The crude total incidence of CTR in the population over the age of 20 per 100,000 was 32.2 (29.6–34.8, 95% CI).

4.Number of surgeries by prefecture per population.

The average number of annual operations from 2014 to 2019 is compared by prefecture in Figure 4. OCTR was performed in Kumamoto and Shimane in more than 50 cases per 100,000 people. This was followed by Oita, Akita, and Nagano. The number of ECTRs was highest in Kochi, Saga, Aomori, and Okayama in that order, each with more than 20 cases per 100,000 people. The age standardized incidence ratio of OCTR was highest in Kumamoto, Shimane, and Oita, while it was highest for ECTR in Kochi, Saga, and Okayama. There was a mild correlation coefficient for ECTRs and the number of hand surgery specialists by prefecture per population (*r* = 0.32, *p* = 0.04). However, the number of hand surgery specialists by prefecture and OCTR per population was not correlated (*r* = 0.06, *p* = 0.67) (Appendix A). 

5.Trends in outpatient and inpatient surgeries.

The number of outpatient and inpatient surgeries for the 5 years (2015–2019) during which data were collected was compared by the procedure. Outpatient surgery was more common for both. ECTR had a higher proportion of outpatient procedures, but there was no significant difference (Table 1). Concerning outpatient and inpatient surgery, the former increased for both OCTR and ECTR, as shown in Figure 5.

## 4. Discussion

In the analysis of the NDB-ODJ, we found that nearly 40,000 CTRs were performed annually in Japan. According to the Statistics Bureau of Japan, in 2019 [17], among the general population of persons aged 20 years or older, the crude annual incidence of CTR was 32.2 (29.6–34.8, 95% CI) per 100,000. It reflected a smaller incidence of CTR compared to studies from the United States and Canada which showed the average annual incidence to be 100 to 300 per 100,000 [7,8]. As shown in Figure 3c, the WHO World Standard Population adjusted annual incidence of CTR is higher than the crude annual incidence. The WHO World Standard Population has a lower percentage of elderly people compared to the Japanese age distribution [18]. Therefore, the adjusted annual incidence of CTR became higher. In Sweden, the incidence of first-time CTR was reported to be 151 in women and 65 in men per 100,000 [19]. It is not clear why the CTR numbers in Japan are smaller than Europe and the United States. One of the reasons might be that the rate of obesity is much lower in Japan. According to the 2017 OECD Obesity Update, obesity rates of adults in Japan were the lowest (4.2%) among OECD countries. Obesity rates in the United States, Canada, and Sweden were 40% (highest), 28.1%, and 13%, respectively [20]. However, the number of CTS patients in Japan is expected to rise. Accumulating evidence from the present study and elsewhere suggests that longer lifespans and increasing rates of diabetic morbidity are negatively impacting the incidence of CTS [6].

We determined that the ratio of OCTR to ECTR was approximately 3:1 in Japan. This reflects a higher prevalence of ECTR compared to the United States where, according to a nationwide study by Foster et al. of a 5-year period from 2007 to 2011, ECTR was only performed 16.1% of the time, with the majority (83.9%) being OCTR [21]. During the period, while OCTR was predominant, ECTR increased significantly as a share of all procedures from 14.0% to 18.8%. A more recent report drawing upon a large subset within the same database, but extended through 2014, showed that ECTR was performed even less—about 15% of the time. This dataset, from one of the largest private medical insurance companies in the U.S., revealed that the number of both the ECTR and OCTR procedures increased from 2007 to 2014 [22]. In our investigation, there was no significant change in the proportion of surgical procedures between 2014 and 2019.

More female patients aged 75–79 underwent CTR, while the largest percentages of male patients were in the 65–69 range for OCTR and 70–74 for ECTR. Surprisingly, the incidence of OCTR was highest in the range of 80–84 years for both females and males, while for ECTR, the incidence was highest at 80–84 for females and 75–79 for males. The age difference at the time of the surgical procedure was likely associated with the average life expectancy in Japan, which was 87 years for women and 81 years for men in 2019 [23]. Successful outcomes following CTR have been reported even in the elderly, and surgery is performed irrespective of age if it is indicated and desired by the patient [24,25]. 

In terms of the geographical distribution of surgeries, both OCTR and ECTR were performed more often in rural than in urban areas. The percentage of elderly people in these areas might have had an effect. In fact, Akita, Kochi, and Shimane—the three prefectures with the highest proportions of elderly [26]—had the greatest numbers of surgeries per capita. Even after standardization by age, there was a tendency for OCTR to be performed more often in rural areas. On the other hand, the fact that the number of ECTRs differed by prefecture might have been due to the influence of the surgeons themselves. The number of ECTRs per population can vary considerably due to the proximity, ability, and predilection of surgeons who actively perform endoscopic surgery in the region. There was a mild correlation coefficient between the number of ECTRs and hand surgery specialists by prefecture per population (*r* = 0.32, *p* = 0.04). However, the number of hand surgeons per capita by region and OCTR per capita was not correlated (*r* = 0.06, *p* = 0.67). Reports from Sweden, Italy, and the United States also show regional variation in the number of surgeries per capita, with occupational factors and access to specialist care having a substantial effect [19,27,28]. 

The ratio of inpatient to outpatient surgeries was about 1:2. ECTR was performed on an outpatient basis more often than OCTR, likely due to it being less surgically invasive. While outpatient surgery is increasingly performed in Japan, the rates are considerably different compared to the United States where, by 2006, more than 99% of CTRs were performed in an ambulatory setting [9]. In Japan, it is not unusual for admission to a hospital to be available for a medical or surgical treatment that might typically be conducted on an outpatient basis in other countries. Comparatively, there is ample capacity—Japan and South Korea lead OECD countries in the number of beds per capita—and the average length of hospitalizations is long [29]. Despite this, healthcare expenditures per GDP are relatively low. With respect to both type and setting, the cost of operative treatment for CTS remains moderate in Japan’s tightly regulated medical system, and the outcome is typically favorable. Nevertheless, considering all factors and given the increasing economic pressure, it is worth analyzing and debating whether the continuation of the current trend related to inpatient surgery for CTS remains an effective allocation of limited medical resources.

This study has several limitations. The NDB-ODJ information on which our findings are based did not include statistics on either the treatment results or complications. In the NDB-ODJ, ECTR is identified by a specific code, but OCTR might have been labeled “neurolysis”, which has another code (K188) for some conditions such as secondary or recurrent CTS. Nevertheless, this study identifies recent trends in CTR within Japan and serves as a reference point with which to understand and optimize treatments.

In the future, by combining the NDB open data with the medical record information including treatment results, even more comprehensive CTS and CTR trends will be elucidated.

## 5. Conclusions

We analyzed nationwide trends in open and endoscopic CTR using comprehensive open data maintained by the Ministry of Health, Labour and Welfare of Japan. Nearly 40,000 CTRs are performed annually in Japan, and in recent years, OCTR has been performed as a surgical treatment about three times more frequently than ECTR. The crude annual incidence of CTR was 32.2 (29.6–34.8, 95% CI) per 100,000, which is lower than Europe and the United States. By age group, both men and women underwent surgery most often in their 70s, and the annual incidence of surgery was highest in those who were 80–84 years old. In terms of the population ratio, there was a tendency for surgery to be performed more often in rural areas than in urban areas. During the study period, outpatient surgery increased compared to inpatient surgery, and was approximately twice as common. Nevertheless, inpatient surgery is still relied upon more frequently in Japan than in other countries. The findings from this study will help to develop future healthcare strategies for CTS.

## Figures and Tables

**Figure 1 jcm-11-04966-f001:**
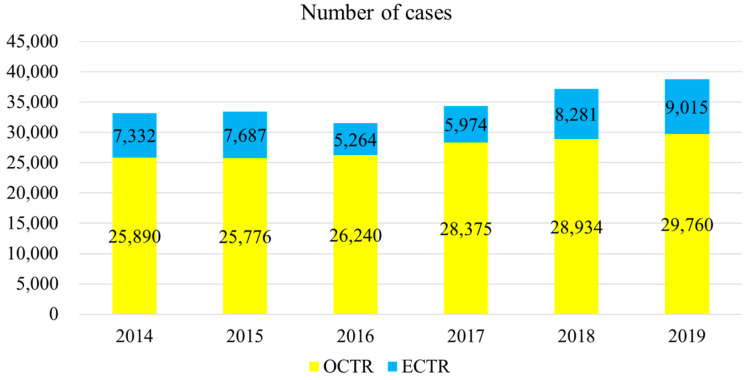
Trends in surgical techniques, 2014–2019 (6 years). OCTR; open carpal tunnel release, ECTR; endoscopic carpal tunnel release.

**Figure 2 jcm-11-04966-f002:**
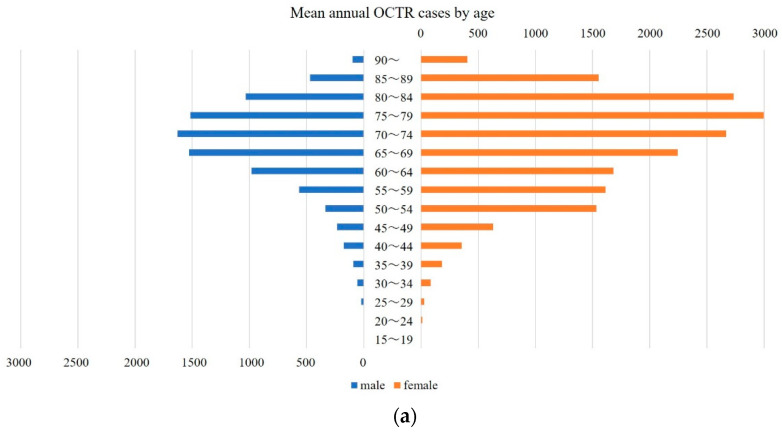
Mean age distribution by type of surgery and gender. Mean annual open carpal tunnel release (OCTR) (**a**) and endoscopic carpal tunnel release (ECTR) (**b**) cases by age.

**Figure 3 jcm-11-04966-f003:**
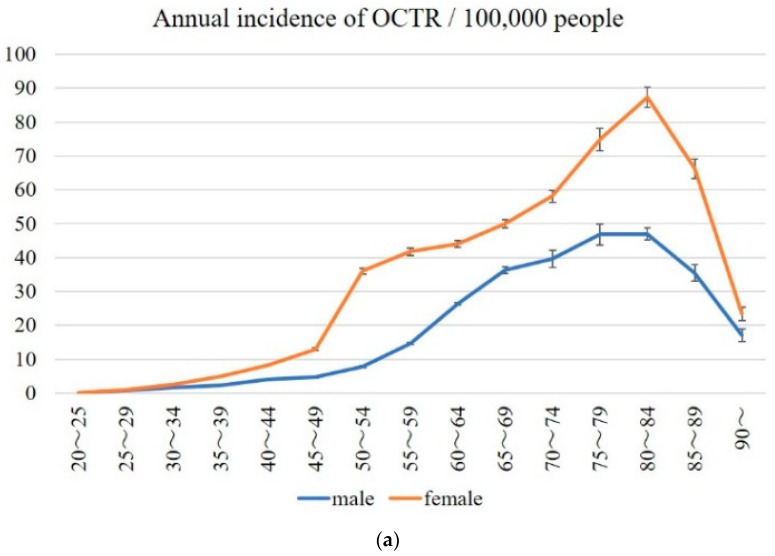
Annual incidence of OCTR and ECTR. Mean annual incidence of OCTR (**a**), ECTR (**b**), and WHO World Standard Population adjustment and crude annual incidence of CTR (**c**) per 100,000 people. OCTR; open carpal tunnel release, ECTR; endoscopic carpal tunnel release, CTR; carpal tunnel release.

**Figure 4 jcm-11-04966-f004:**
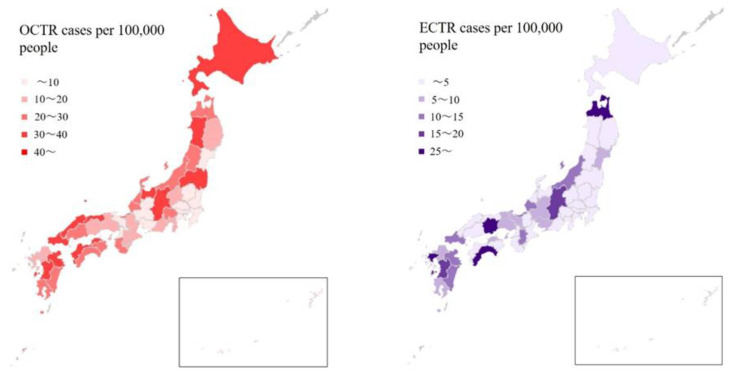
Mean annual open carpal tunnel release (OCTR) and endoscopic carpal tunnel release (ECTR) cases per 100,000 people by prefecture.

**Figure 5 jcm-11-04966-f005:**
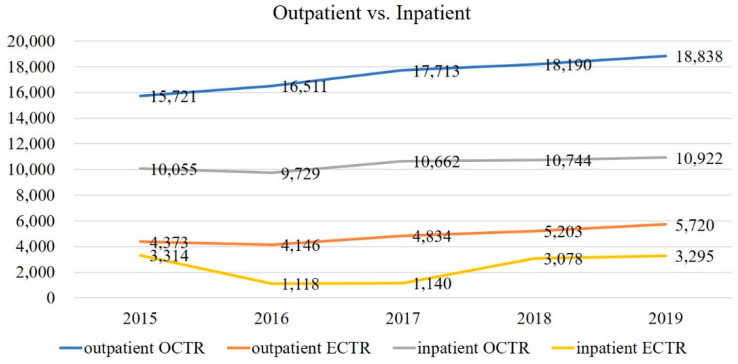
Trends in outpatient and inpatient surgeries, 2015–2019 (5 years). OCTR; open carpal tunnel release, ECTR; endoscopic carpal tunnel release.

**Table 1 jcm-11-04966-t001:** Comparison of outpatient and inpatient surgery by surgical procedure, 2015–2019 (5 years).

Setting	OCTR	ECTR	*p* Value
Outpatient	86,972	24,276	0.11
Inpatient	52,112	11,945	

OCTR; open carpal tunnel release, ECTR; endoscopic carpal tunnel release.

## Data Availability

Internet links to the Ministry of Health, Labour and Welfare of Japan datasets used in this study are provided in the References section.

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
