# Peer review of "Trends in Open vs. Endoscopic Carpal Tunnel Release: A Comprehensive Survey in Japan"

_jcm, 2022, doi:10.3390/jcm11174966_

Round 1

Reviewer 1 Report

The authors have addressed the issues raised in the previus review.

Author Response

Thank you for supporting comments.

Reviewer 2 Report

Dear Authors,

thank you for your responses and the edited manuscript. The figures look better now, and the manuscript has improved in general. Still, my concern on age-standardization of the results remains. I'm sorry for not being precise enough with my previous comments on it: I think you should use an Asian standard population if you have one (?), and if there's not an Asian population as a reference population, maybe you could use the WHO standard population. Now, the results are standardized for Japanese population in 2019, which unfortunately brings no value for international comparison. Also, the method of standardization (directs/indirect/other) must be mentioned. This really would bring valuable data on your research!

Warmly,

your reviewer

Author Response

Unlike the European standard population, there is not an Asian standard reference population. Therefore, for comparative purposes, we performed age-specific population rate adjustments using the WHO World Standard Population. It was done by using a direct approach.

The following explanation was added to the Materials and Methods section:

The WHO World Standard Population distribution was used to make age-based international comparisons. Calculations using a direct approach were performed to adjust for age at carpal tunnel release, the Japanese population in 2019, and crude mean annual incidence of CTR for 6 years. 

Further, in the Results section, we recreated the following figure:

Figure 3c. WHO World Standard Population adjustment and crude annual incidence of CTR per 100,000 people.

The following was appended to the Discussion section.

As shown in Figure 3c, the WHO World Standard Population adjusted annual incidence of CTR is higher than the crude annual incidence. The WHO World Standard Population has a lower percentage of elderly people compared to the Japanese age distribution. Therefore, the adjusted annual incidence of CTR became higher.

We also added the following citation to document the method:

Reference 18. Ahmad, O.B.; Boschi-Pinto, C; Lopez, A.D.; Murray, C.J.L.; Lozano, R.; Inoue, M. Age standardization of rates: a new WHO standard. Geneva: World Health Organization, 2001.

This manuscript is a resubmission of an earlier submission. The following is a list of the peer review reports and author responses from that submission.

Round 1

Reviewer 1 Report

This is a relatively simple study based on official registry data apparently available online. Some of the information would be of interest to readers and researchers within the field.

Introduction

L 35: the statement about risk of nerve damage needs to be rephrased, it is what the authors of that referenced article stated rather than a proven fact.

The segment about cells and elderly is irrelevant for the subject of this article and better removed.

What are the hypotheses based on? This segment may not be necessary for this type of study that simply reports downloadable data from an open registry. Stating the aims is sufficient.

Materials and Methods

Does the registry specify if the surgeon was a hand surgery specialist? If not, this analysis is not valid. The experience in a procedure does not necessarily depend on whether the surgeon is a hand or orthopedic surgeon. This applies specially to carpal tunnel release.

Results

Figure 1: percent is more informative than numbers, should be added to text or figure.

How did you statistically test “trends in surgical procedures” over the years (p-value)?

Figures 2:

In general, number of cases or procedures is not adequately informative, this type of data should be shown as incidence rates and comparisons as incidence rate ratios with 95% confidence intervals.

Figure 3: This figure showing number of cases is probably unnecessary because Figure 4 showing incidence rates is more informative.

Different regions may differ in demographics, the appropriate way is to compare incidence rates adjusted for age and sex.

It is astonishing that carpal tunnel release surgery is still done as inpatient in so many cases. This information is certainly not generalizable to other parts of the world and needs to be explained much better in the Discussion. In fact it should be a subject for a national debate about how resources can be used in this way.

Discussion

L 160: US figures 2007-2011 are old, there should be more recent data for comparison.

The age distribution for women is very different from studies from previous studies from other countries, need to expand this.

Author Response

REVIEWER

This is a relatively simple study based on official registry data apparently available online. Some of the information would be of interest to readers and researchers within the field.

Introduction

L 35: the statement about risk of nerve damage needs to be rephrased, it is what the authors of that referenced article stated rather than a proven fact.

RESPONSE

Thank you, we accept that insight. We removed the sentence entirely and replaced it with the following:

A common feature of both open and endoscopic carpal tunnel release is to divide the transverse carpal ligament, although each technique has its own advantages and disadvantages.

REVIEWER

The segment about cells and elderly is irrelevant for the subject of this article and better removed.

RESPONSE

This section, beginning around Line 45, attempts to explain why reporting the data on Japanese carpal tunnel surgery is particularly valuable. It states that the incidence of carpal tunnel syndrome increases with aging, and then highlights a subject emerging in recent research—that senescence cells are involved in the onset. Therefore, we think that the trends associated with carpal tunnel surgery in Japan, the leading super-aged society, will be useful for countries that will be facing similar trajectories of managing their own super-aging societies in the future as well as those involved in such research.

We would prefer to retain this section and hope we have made a reasonable argument for it.

REVIEWER

What are the hypotheses based on? This segment may not be necessary for this type of study that simply reports downloadable data from an open registry. Stating the aims is sufficient.

RESPONSE

Good point. That makes sense to us. we removed hypothesis section in its entirety.

REVIEWER

Materials and Methods

Does the registry specify if the surgeon was a hand surgery specialist? If not, this analysis is not valid. The experience in a procedure does not necessarily depend on whether the surgeon is a hand or orthopedic surgeon. This applies specially to carpal tunnel release.

RESPONSE

The NDB registry does not indicate whether the surgery was performed by a hand surgeon or an orthopedic surgeon. The number of hand surgeons in each region is examined separately, and we found that there was a correlation between the number of hand surgeons per capita and the number of ECTRs per capita by region (r = 0.32, p = 0.04). On the other hand, the number of hand surgeons per capita by region and OCTR per capita did not have a correlation (r = 0.06, p = 0.67). This suggests that ECTR was mainly performed by hand surgeons, and OCTR was performed by non-hand surgery specialists as well. We added these findings using similar language to the Results and Discussion sections.

REVIEWER

Results

Figure 1: percent is more informative than numbers, should be added to text or figure.

RESPONSE

We added the percent of ECTR from 2014 to 2019 to the text (it appears just before the figure placement). It reads:

From 2014 to 2019, the ECTR percentages were 28%, 30%, 20%, 21%, 29% and 30%, respectively, and the overall percentage was 26%.

REVIEWER

How did you statistically test “trends in surgical procedures” over the years (p-value)?

RESPONSE

An m × n contingency table was used to determine statistical differences in procedures by year.

REVIEWER

Figures 2:

In general, number of cases or procedures is not adequately informative, this type of data should be shown as incidence rates and comparisons as incidence rate ratios with 95% confidence intervals.

RESPONSE

We added a new Figure 3, entitled "Mean annual incidence of OCTR and ECTR per 10,000 people," to explain the annual incidence of OCTR and ECTR standardized by age and sex. We also provided the incidence with 95% confidence intervals.

REVIEWER

Figure 3: This figure showing number of cases is probably unnecessary because Figure 4 showing incidence rates is more informative.

RESPONSE

We deleted Figure 3.

REVIEWER

Different regions may differ in demographics, the appropriate way is to compare incidence rates adjusted for age and sex.

RESPONSE

We calculated the age standardized incidence ratios of OCTR and ECTR using the mean number of cases in each prefecture and a 2019 population estimation summary. The results are written in Appendix 1.

REVIEWER

It is astonishing that carpal tunnel release surgery is still done as inpatient in so many cases. This information is certainly not generalizable to other parts of the world and needs to be explained much better in the Discussion. In fact it should be a subject for a national debate about how resources can be used in this way.

RESPONSE

Under the universal health insurance system in Japan, it is not unusual for hospitalization to be available for many forms surgical treatment as well as other medical care. Along the lines of your observation, the number of hospital beds exceeds those required number nationwide, and compared internationally, the number of beds per capita is large, the average length of hospital stay is long, and the number of medical staff per bed is small.

We added this sentence in the Discussion section:

In Japan, it is not unusual for admission to a hospital to be available for a medical or surgical treatment that might typically be conducted on an outpatient basis in other countries. Comparatively, there is ample capacity—Japan and South Korea lead OECD countries in the number of beds per capita—and the average length of hospitalizations is long [30]. Despite this, healthcare expenditures per GDP, are relatively low. With respect to both type and setting, the cost of operative treatment for CTS remains moderate in Japan's tightly regulated medical system, and the outcome is typically favorable. Neverthless, considering all factors in light of increasing economic pressure, it is worth analyzing and debating whether further sustaining the trend related to inpatient surgery for CTS remains an effective allocation of limited medical resources.

REVIEWER

Discussion

L 160: US figures 2007-2011 are old, there should be more recent data for comparison.

RESPONSE

A more recent report drawing upon a large subset within the same database but extending it through 2014 showed that ECTR was performed even less—about 15% of the time. This dataset, from one of the largest private medical insurance companies in the U.S., revealed an increase in in the numbers of both ECTR and OCTR from 2007–2014.

REVIEWER

The age distribution for women is very different from studies from previous studies from other countries, need to expand this.

RESPONSE

We added annual incidence of OCTR and ECTR per 10,000 people in Figure 3. Accompanying that addressing your insight is the following

Surprisingly, the incidence of OCTR was highest in the range of 80–84 years for both females and males; while for ECTR, the incidence was highest at 80–84 in females and 75–79 in males.

Reviewer 2 Report

Dear authors,

thank you for your manuscript " Trends in Open vs Endoscopic Carpal Tunnel Release: A Comprehensive Survey in Japan"! It was interesting to read. However, I have some comments I think would improve your work. My main concern here is that the results are not standardized. I think standardizing them would make them comparable internationally (not only nationally as they are at the moment). I hope my comments help you to improve this important research!

Warmly,

your reviewer

Author Response

REVIEWER

Dear authors,

thank you for your manuscript " Trends in Open vs Endoscopic Carpal Tunnel Release: A Comprehensive Survey in Japan"! It was interesting to read. However, I have some comments I think would improve your work. My main concern here is that the results are not standardized. I think standardizing them would make them comparable internationally (not only nationally as they are at the moment). I hope my comments help you to improve this important research!

RESPONSE

Thank you so much for your comments. We added the age and sex-specific incidence of OCTR and ECTR as Figure 3. Mean annual incidence of OCTR and ECTR per 10,000 people. We also added the data on the standardized incidence ratio of OCTR and ECTR in each prefecture in Appendix 1. Prefectural data. Those data offer a point of comparison to those of other countries.